# Ca^2+^ Homeostasis by Plasma Membrane Ca^2+^ ATPase (PMCA) 1 Is Essential for the Development of DP Thymocytes

**DOI:** 10.3390/ijms24021442

**Published:** 2023-01-11

**Authors:** David Beckmann, Kristina Langnaese, Anna Gottfried, Johannes Hradsky, Kerry Tedford, Nikhil Tiwari, Ulrich Thomas, Klaus-Dieter Fischer, Mark Korthals

**Affiliations:** 1Institute for Biochemistry and Cell Biology, Medical Faculty, Otto-von-Guericke-University Magdeburg, 39120 Magdeburg, Germany; 2Department of Cellular Neuroscience, Leibniz Institute for Neurobiology, 39120 Magdeburg, Germany

**Keywords:** PMCA1, PMCA4, Ca^2+^ homeostasis, T cell development, thymocytes, DN to DP transition, TCR signaling, Notch1

## Abstract

The strength of Ca^2+^ signaling is a hallmark of T cell activation, yet the role of Ca^2+^ homeostasis in developing T cells before expressing a mature T cell receptor is poorly understood. We aimed to unveil specific functions of the two plasma membrane Ca^2+^ ATPases expressed in T cells, PMCA1 and PMCA4. On a transcriptional and protein level we found that PMCA4 was expressed at low levels in CD4^−^CD8^−^ double negative (DN) thymocytes and was even downregulated in subsequent stages while PMCA1 was present throughout development and upregulated in CD4^+^CD8^+^ double positive (DP) thymocytes. Mice with a targeted deletion of *Pmca1* in DN3 thymocytes had an almost complete block of DP thymocyte development with an accumulation of DN4 thymocytes but severely reduced numbers of CD8^+^ immature single positive (ISP) thymocytes. The DN4 thymocytes of these mice showed strongly elevated basal cytosolic Ca^2+^ levels and a pre-mature CD5 expression, but in contrast to the DP thymocytes they were only mildly prone to apoptosis. Surprisingly, mice with a germline deletion of *Pmca4* did not show any signs of altered progression through the developmental thymocyte stages, nor altered Ca^2+^ homeostasis throughout this process. PMCA1 is, therefore, non-redundant in keeping cellular Ca^2+^ levels low in the early thymocyte development required for the DN to DP transition.

## 1. Introduction

The development of T cells is a tightly controlled process that guarantees the tolerance against self-antigens but also allows for mounting an antigen-specific immune reaction when it is required [1]. Before the expression of a T cell receptor (TCR), the survival and proliferation of early CD4^−^CD8^−^ double negative (DN) thymocytes are mainly driven by Notch1 and IL-7 signaling [2]. At the CD25^+^CD44^low^ DN3 stage, a pre-TCR is built from the newly recombined β-chain and an invariant pre-TCR α (pTα)-chain. During β-selection, the functional recombination of the β-chain is proven by intrinsic signaling of the pre-TCR required for a second wave of proliferation and differentiation via a CD8^+^ immature single positive (ISP) stage into CD4^+^CD8^+^ double positive (DP) thymocytes [3]. Upon recombination of the TCR-α chain in DP thymocytes, the mature αβ-TCR is formed. In the course of positive selection, only those DP thymocytes survive and differentiate into CD4^+^ or CD8^+^ single positive (SP) thymocytes which express an αβ-TCR capable of recognizing peptide-MHC complexes on thymic stromal cells, while all others die by neglect. During negative selection, the DP and SP thymocytes which receive strong TCR signals upon recognition of self-peptides are also eliminated by apoptosis. As a consequence, only functional and self-tolerant SP thymocytes finally exit the thymus [1].

Spatio-temporal variations in the intracellular Ca^2+^ concentration are particularly important in determining the TCR signal strength that decides over survival, proliferation and differentiation [4]. TCR stimulation triggers the activation of PLC-γ, the production of inositol 1,4,5-trisphosphate (IP3) and the release of Ca^2+^ from the ER into the cytosol through the IP3 receptor. T cells are capable of amplifying cytosolic Ca^2+^ signals by store-operated Ca^2+^ entry (SOCE) which is induced by the interaction of the ER Ca^2+^ sensor stromal interaction molecules (STIM) 1 and 2 and the plasma membrane Ca^2+^ channels ORAI 1-3 [5]. SOCE is critical for activating the nuclear factor of activated T cells (NFAT), cytokine production, metabolic reprogramming and the proliferation of T cells. Mutations in STIM and ORAI isoforms may result in severe combined immune deficiency (SCID) in mice and humans [6,7,8,9,10]. Furthermore, STIM1/2 double deficient mice also show lymphoproliferative disease probably due to the impaired development of regulatory T cells (Tregs) [8,9]. 

Although SOCE is surprisingly not required for the development of conventional αβ-T cells [8], Ca^2+^ signaling is important during thymocyte selection. For example, the conditional deletion of PLC-γ markedly reduced the numbers of SP thymocytes in mice due to defective positive and negative selection [11,12]. Two-photon live cell imaging revealed that TCR-mediated Ca^2+^ oscillations limit the motility of DP thymocytes in order to prolong the interaction with antigen-presenting stromal cells and to warrant sufficient signals required for positive selection [13]. Setting the signaling threshold for positive or negative selection has been attributed to the balance of Ca^2+^ downstream pathways leading to the activation of Calcineurin/NFAT [14,15] or of Ca^2+^/Calmodulin-dependent protein kinase (CaMK) type IV/Gr and PKC [16], respectively. Insights into Ca^2+^ signaling in early thymocyte development, however, are more limited. For example, mice lacking all three IP3 receptor isoforms had a partial block of DP thymocyte development at the DN4 and ISP stages and developed T cell acute lymphoblastic leukemia [17]. Mice deficient for other Ca^2+^ channels, such as the voltage-gated Ca^2+^ (Ca_V_) channels Ca_V_1.2 and Ca_V_1.3 [18] or TRPM7 [19], or lacking the Ca^2+^ downstream effector Calcineurin B1 [15,20], showed a developmental block at the DN3 stage due to impaired β-selection. 

The family of plasma membrane Ca^2+^ ATPases (PMCA) comprises four isoforms, each with several alternatively-spliced variants [21]. PMCA2 and 3 are mainly expressed in excitable cells such as neurons and inner ear hair cells [22,23,24,25], and mutations in PMCA2 and 3, thus, can cause deafness or ataxia in mice and humans [26,27,28,29]. PMCA1 and 4 are ubiquitously expressed, particularly in non-excitable cells such as immune cells [30]. PMCAs are mainly activated by Ca^2+^/Calmodulin besides other mechanisms and their function and stability is dependent on the interaction with Neuroplastin and/or Basigin [31,32,33,34]. PMCA4 is important for sperm motility. Its deficiency in mice thus causes male sterility but has no impact on development [35,36]. PMCA1, on the other hand, has been regarded as a housekeeping enzyme as the deletion of its gene in mice is embryonically lethal [35]. It is required for the naïve exit of stem cells from pluripotency [37]. Recently, PMCA1 mutations have been shown to cause a monogenic form of neurodevelopmental disorder in humans [38]. In addition, reduced expression of PMCA1 has been shown to be a risk factor for hypertension and associated disease [39]. 

PMCA1 and PMCA4 are the only isoforms expressed in T cells and both are upregulated in stimulated T cells [33,40,41]. Several studies in Jurkat T cells and primary human T cells have shown that PMCA4 tunes TCR-induced SOCE within certain microdomains of the T cell immune synapse [30,40,42,43,44,45]. There, it acts both as a negative or as a positive regulator of SOCE and NFAT activation depending on its exact positioning. STIM1 interaction with PMCA4 [45] and Ca^2+^ buffering by mitochondria [44] both limit the Ca^2+^-induced PMCA4 activity while PMCA4 interaction with partner of STIM (POST) [43] protects PMCA4 from STIM-mediated inhibition, thereby promoting Ca^2+^ extrusion in the vicinity of ORAI in order to attenuate the Ca^2+^-dependent inactivation of ORAI [42]. 

Although the importance of PMCA4 in modulating SOCE in T cells is clear, the roles of PMCA1 in lymphocyte signaling and PMCA functions during T cell development, where SOCE seems to play a minor role, are largely unknown. We have previously reported that PMCA1 but not PMCA4 is required for normal B cell development [46]. In the present study, we expand this knowledge and show that PMCA4 is also dispensable for normal T cell development and thymocyte Ca^2+^ homeostasis in mice, whereas PMCA1 has a non-redundant role in early thymocyte development with a specific requirement for the transition from DN4 to ISP thymocytes.

## 2. Results

### 2.1. Differential Expression of PMCA1 and PMCA4 during Thymocyte Development

A role for PMCA4 in T cell biology has mainly been demonstrated either in human T cell lines or in activated primary T cells. Based on our previous finding that PMCA1 plays a dominant role in B cell development [46], we were curious about the impact of both isoforms on T cell development. First, we looked at the regular thymic expression of PMCA1 and PMCA4 in wildtype mice. Immunoblots of total thymocyte protein extracts revealed a robust expression of PMCA1 (Figure 1A) while PMCA4 was present but hardly detectable (Figure 1C). ImmGen RNA expression data (www.immgen.org accessed on 7 July 2022) [47] suggest that PMCA4 and PMCA1 are differentially regulated during thymocyte development. We therefore analyzed the PMCA expression in DN thymocytes compared to the total thymocytes that comprise more than 80% of DP thymocytes. Equal protein amounts from enriched wildtype DN thymocyte extracts showed lower PMCA1 (Figure 1A,B) but more robust PMCA4 amounts (Figure 1C,D), compared to the total thymocyte extracts. These results thus confirm the RNA expression data that PMCA4 is expressed very early during the DN stage but is then strongly downregulated during the DN–DP transition while PMCA1, which is also expressed at the early stages, is upregulated upon the DN–DP transition.

Since a direct juxtaposition of PMCA1 and PMCA4 antibody signals is not suitable to compare the expression levels of both isoforms, we took advantage of a previously published RT-PCR-based method to quantify the relative *Pmca1* and *Pmca4* mRNA expression using primers common to both cDNAs and two restriction enzymes with each specifically cutting either the *Pmca1* or the *Pmca4* cDNA, respectively (Figure 1E) [35,46]. The PCR results in the samples of total wildtype thymocytes showed no or only faint *Pmca4* cleavage product but an equal band intensity of the uncut cDNA when using the *Pmca4*-specific restriction enzyme (BglI) compared to the control without enzymes (Figure 1F). In turn, clear *Pmca1* cleavage products but no or only faint uncut cDNA were visible when using the *Pmca1*-specific enzyme (ClaI, Figure 1F). We observed similar results in the wildtype DN thymocyte samples with the exception that the *Pmca4* cDNA was more reliably detected than in the total thymocytes but still with a much lower band intensity compared to bands specific for *Pmca1* (Figure 1F). In the control samples from wildtype brain or spleen tissue, respectively, both *Pmca* cDNAs were readily detectable with the brain showing an almost equal ratio of band intensities and the spleen also showing less intensity for *Pmca4* (Figure 1F). Therefore, relative *Pmca1* and *Pmca4* cDNA detection confirmed the impression from the immunoblots that PMCA1 is the predominant PMCA isoform not only in DP thymocytes where PMCA4 is strongly downregulated but also in DN thymocytes, where PMCA4 expression is higher.

### 2.2. PMCA1 but Not PMCA4 Is Required for Normal Thymocyte Development

Hence, we speculated that a lack of PMCA4 or PMCA1 would have different effects on thymocyte development. In order to test this, we analyzed mice with a germline deletion of the *Pmca4* gene and mice with a conditional deletion of the *Pmca1* gene under the control of the pTα promoter (*Pmca1^f/f^ pTα-Cre^+^*), respectively. To our surprise, the thymi of *Pmca4^−/−^* mice did not show any alterations in terms of frequencies and absolute numbers of DN, DP and SP thymocytes (Figure 2A,B), as well as DN subpopulations (Figure 2C,D), compared to the *Pmca4^+/+^* control mice. As we have demonstrated before in mice on an FVB/N background [46], peripheral T and B cell numbers in the spleen and lymph nodes of *Pmca4^−/−^* mice were unaltered (Appendix A). Additionally, naïve T cells were present in normal numbers (Appendix A). Moreover, γδ-T cells, NKT cells as well as regulatory T cells (Tregs) developed normally in the thymi of *Pmca4^−/−^* mice (Appendix A).

In stark contrast to the *Pmca4^−/−^* mice, we found that the thymi of *Pmca1^f/f^ pTα-Cre^+^* mice were much smaller with drastically reduced cell numbers than those of the *Pmca1^f/f^* control mice. CD4/CD8 staining showed that almost all of these cells were DN thymocytes (Figure 2E). The DP population was substantially atrophied although SP thymocytes appeared as prominent populations. Of note, these SP cells showed abnormally high expression levels of CD44 compared to the DP thymocytes and SP thymocytes of control mice (Appendix A). However, an absolute count revealed that the DN thymocyte numbers in *Pmca1^f/f^ pTα-Cre^+^* mice were even increased, while the numbers of all other populations were severely reduced compared to the control mice (Figure 2F). Particularly, there was an increasing accumulation of DN3 and DN4 thymocytes, respectively (Figure 2G,H). Wildtype CD8^+^ SP thymocytes also comprise the TCR^low^ CD8^+^ ISP population that derives from DN4 thymocytes before becoming DP. Similar to DP thymocytes, this population was also barely detectable in the *Pmca1^f/f^ pTα-Cre^+^* mice (Figure 2I,J), which, therefore, indicates that the collapse of thymic cell numbers had already started before the DP formation. As a control, the development of γδ-T cells, that probably diverge before the establishment of pTα-driven PMCA1 deficiency was operational, and γδ-T cell numbers were even increased in the *Pmca1^f/f^ pTα-Cre^+^* mice (Appendix A). Interestingly, the total leukocyte counts in spleen and lymph node samples of the*Pmca1^f/f^ pTα-Cre^+^* mice were not significantly different compared to the control mice. The CD8+ and, to a lesser extent, CD4^+^ T cell numbers, however, were significantly reduced in both organs (Appendix A). CD62L/CD44 staining revealed an almost complete lack of both naïve CD4^+^ and naïve CD8^+^ T cells in the spleen and lymph nodes (Appendix A), which may indicate homeostatic proliferation of a few thymic emigrants in the *Pmca1^f/f^ pTα-Cre^+^* mice that were able to complete the development in an otherwise lymphopenic environment. Although most *Pmca1^f/f^ pTα-Cre^+^* mice appeared generally healthy, approximately one-third showed a worsened general condition marked by a smaller size and increased incidence of rectal prolapses or other sporadic inflammatory conditions. T cells in these mice were, therefore, not fully capable of controlling inflammation. 

We were wondering if indeed some PMCA1-deficient thymocytes could overcome the critical step at the DN–DP transition capable of filling the peripheral T cell pool. First, we tracked the Cre expression under the pTα promoter in *Pmca1^f/f^ pTα-Cre^+^* mice that also express a membrane-targeted, two-color fluorescent tomato/GFP (*mT/mG*) reporter allele, that upon Cre activity switches from a constitutive production of red fluorescence to green fluorescence. As expected, the GFP expression in these mice was first prominently visible merely at all DN3 thymocytes and was highest in DN4 (Appendix A); however, GFP was only detectable in a very small fraction of the few present DP and SP thymocytes, respectively (Appendix A). We therefore concluded that most thymocytes that developed beyond the DN stage did not efficiently express Cre and, thus, escaped *Pmca1* deletion. Indeed, PMCA1 protein was detectable in equal amounts in CD4^+^ T cell lysates from the control and *Pmca1^f/f^ pTα-Cre^+^* mice, while heterozygous *Pmca1* deletion in the Cre-expressing mice with only one floxed allele (*Pmca1^f/+^ pTα-Cre^+^*) resulted in a PMCA1 protein reduction by approximately half (Appendix A). Because the PMCA1 protein levels might be upregulated in wildtype CD44^high^ T cells, a conclusion on the relative proportion of wildtype cells is still difficult. We therefore performed a genomic PCR detecting all three, namely, the wildtype (wt), the floxed (f uncut) and the Cre-deleted (f cut) *Pmca1* alleles (Appendix A). We found only weak bands of the Cre-deleted allele in T cells from the *Pmca1^f/f^ pTα-Cre^+^* mice, e.g., with an intensity much lower than that detected in the T cells from heterozygous mice which did not show any uncut floxed DNA and so were expected to contain 50% Cre-deleted DNA (Appendix A). In conclusion, only very few T cells with *Pmca1* deletion (either homozygous or heterozygous) added to the peripheral T cell pool of the *Pmca1^f/f^ pTα-Cre^+^* mice. 

Taken together, while PMCA1 expression was a strict condition for ISP and DP thymocyte formation and subsequent maturation to naïve T cells, PMCA4 was not evidently required for normal thymocyte development. 

### 2.3. Pmca1 but Not Pmca4 Controls Ca^2+^ Homeostasis during Thymocyte Development

Given the low expression of PMCA4 and its redundancy in T cell development, we wondered if it nonetheless does have an impact on thymocyte Ca^2+^ homeostasis. The thymocytes of *Pmca4^−/−^* mice did not show significantly different basal Ca^2+^ levels in the DN, DP and SP populations compared to the control mice (Figure 3A,B). In addition, TCR-induced SOCE was fully operational in all tested thymocyte populations of *Pmca4^−/−^* mice (Figure 3A,C). Impaired Ca^2+^ extrusion from the cytosol across the plasma membrane in PMCA4-deficient thymocytes may be concealed by an enforced buffering of Ca^2+^ in internal stores such as the ER. In order to analyze the ER Ca^2+^ load, we measured the Ca^2+^ leakage from the ER into the cytosol by adding the SERCA blocker, thapsigargin, in a Ca^2+^-free buffer. Basal cytosolic Ca^2+^ levels in the Ca^2+^-free buffer were also equal in all the tested thymocyte populations of *Pmca4^−/−^* mice compared to the control mice (Figure 3D,E). Due to very low cell numbers, the testing of Ca^2+^ levels in very early DN thymocytes such as DN1 and DN2 was not possible. We therefore cannot rule out that the loss of PMCA4 may impact the basal Ca^2+^ levels in these rare early progenitors. However, we did not observe a stronger Ca^2+^ release from the ER upon thapsigargin treatment in any of the thymocyte populations of *Pmca4^−/−^* mice (Figure 3D,F).

Considering that PMCA4 does not obviously contribute to global Ca^2+^ homeostasis in most thymocytes, PMCA1 is probably the only relevant ATPase expelling Ca^2+^ across the plasma membrane in these cells. We therefore analyzed the basal cytosolic Ca^2+^ levels in DN3, DN4 and DP thymocytes of *Pmca1^f/f^ pTα-Cre^+^* mice. While the DN3 thymocytes already had elevated basal Ca^2+^ levels, the DN4 thymocytes showed dramatically increased Ca^2+^ levels compared to the corresponding populations of control mice (Figure 4A,B), which not only confirms that these DN4 thymocytes were PMCA1-deficient but also that the loss of PMCA1 was not perceptibly compensated by other Ca^2+^ expelling mechanisms. Only a minor fraction of the DP thymocytes from *Pmca1^f/f^ pTα-Cre^+^* mice showed higher Ca^2+^ levels (Figure 4A), and a very similar result was also obtained for SP thymocytes (not shown), which again mirrors that most of the very few DP and SP thymocytes in these mice still expressed PMCA1. In conclusion, PMCA1 but not PMCA4 keeps the intracellular Ca^2+^ levels low during all stages of thymocyte development from as early as the DN3 stage. 

### 2.4. Signs of Premature Signaling in PMCA1 Deficient DN4 Thymocytes

Since the recombination of the TCR-β-chain is a prerequisite for passing the β-selection checkpoint [48], and increased Ca^2+^/NFAT signaling might be a potential negative regulator of this process [49], we next asked if the lack of DP thymocytes in *Pmca1^f/f^ pTα-Cre^+^* mice may be due to an impaired β-chain rearrangement. To this end, we crossed *Pmca1^f/f^ pTα-Cre^+^* mice with OT-II TCR transgenic mice that do not require the recombination of the TCR-α- and β-chain. However, the *OT-II^+^ Pmca1^f/f^ pTα-Cre^+^* mice were still unable to produce significant numbers of DP thymocytes (Appendix A). These mice also accumulated DN4 thymocytes (Appendix A) exactly like the non-transgenic *Pmca1^f/f^ pTα-Cre^+^* mice (Figure 2E,F). Therefore, impaired β-recombination, per se, cannot be the primary cause of the strong block at the DN–DP transition. 

Elevated Ca^2+^ levels in the DN3 and DN4 thymocytes of *Pmca1^f/f^ pTα-Cre^+^* may instead transmit signals normally induced by the pre-TCR. In this respect, we looked for additional signs of enhanced signaling. As a typical surrogate marker for TCR activation as well as for pre-TCR activation [50] we first analyzed the CD5 surface expression. Whereas the control thymocytes showed a low CD5 expression in DN3 and a robust upregulation only in a fraction of DN4, the CD5 upregulation in *Pmca1^f/f^ pTα-Cre^+^* mice had already started at DN3 and was high on all DN4 thymocytes (Figure 5A,B). The elevated CD5 expression, indeed, indicated pre-mature TCR signaling in the DN thymocytes of *Pmca1^f/f^ pTα-Cre^+^* mice. Pre-TCR signaling was shown to initiate the downregulation of Notch1 [51]. We therefore also analyzed the cell surface expression of Notch1 as well as its activation status by detecting the cytosolic release of an intracellular Notch1 domain that is cleaved upon the activation of Notch1 [52]. In the control DN thymocytes, Notch1 expression was high on the DN2 and DN3 populations while the DN4 population comprised a Notch1^high^ and a Notch1^low^ fraction (Figure 5C), consistent with published data showing the downregulation of Notch1 at the DN4 stage [51]. This downregulation was accompanied by a decline of its activation status continuously from DN2 to DN4, as measured by the decreasing signal intensity of the cleaved intracellular domain (Figure 5E), and was inversely correlated with the CD5 expression on the thymocytes of control mice (Appendix A). As expected, the Notch1 surface expression and activation status was not significantly altered in the DN2 population of *Pmca1^f/f^ pTα-Cre^+^* mice, but both the Notch1 expression (Figure 5C,D) and signaling (Figure 5E,F) dropped more readily in DN3 and most notably in DN4 below the wildtype levels. Altogether, these results imply that the PMCA1 deficiency and very high basal Ca^2+^ levels promote rather than inhibit β-selection signaling in DN3 and DN4 thymocytes.

### 2.5. DP but Not DN4 Thymocytes of Pmca1^f/f^ pTα-Cre^+^ Mice Are Prone to Apoptosis

We detected massive alterations in Ca^2+^ homeostasis and premature signaling in the DN thymocytes of *Pmca1^f/f^ pTα-Cre^+^* mice, it was unclear though, if the impaired DP thymocyte formation was solely due to an intrinsic defect of DN thymocytes or to the failure of the DP thymocytes to survive. High DN thymocyte numbers and an accumulation of DN4 in *Pmca1^f/f^ pTα-Cre^+^* mice does not point to a proliferation defect prior to DP formation. Indeed, the cell growth of DN thymocytes, as measured by forward scatter characteristics (Appendix A), and the expression of the proliferation marker Ki67 (Appendix A) were not affected. Since the accumulation of intracellular Ca^2+^ is well known to promote the mitochondrial pathway of apoptosis [53], it was obvious to analyze the viability of PMCA1-deficient thymocytes. Surprisingly, the DN thymocytes of *Pmca1^f/f^ pTα-Cre^+^* mice were as viable as the control thymocytes as measured by low 7-AAD and Annexin V staining (Figure 6A,B). On the other hand, the existing DP thymocytes of *Pmca1^f/f^ pTα-Cre^+^* mice were less viable than the DP thymocytes of control mice (Figure 6A,B), even though we did not reveal a significant accumulation of dead DP thymocytes. In addition, using low FSC as a surrogate marker for dead cells (Appendix A), we did not observe an increased frequency of GFP-positive cells within the FSC^low^ dead DP thymocyte population of the Cre-reporter-expressing *Pmca1^f/f^ pTα-Cre^+^* mice (Appendix A). Therefore, we concluded that DP thymocytes, either expressing PMCA1 or not, were equally prone to apoptosis in *Pmca1^f/f^ pTα-Cre^+^* mice. As late apoptotic or dead cells could have been removed by other cells in the intact thymus, we had a closer look at earlier apoptotic markers. We detected slightly, but not significantly, higher intensities of active Caspase 3 staining in the DN3 and DN4 thymocytes of *Pmca1^f/f^ pTα-Cre^+^* mice whereas the DP thymocytes showed a clear upregulation of active Caspase 3 (Figure 6C,D). A very similar result was obtained after staining of p53 that was slightly upregulated in the DN3 and DN4 thymocytes but that was strongly expressed in the DP thymocytes of *Pmca1^f/f^ pTα-Cre^+^* mice compared to the control (Figure 6E,F). Of note, pro-survival factor Bcl-2 was still highly expressed in the DN4 thymocytes of *Pmca1^f/f^ pTα-Cre^+^* mice while it was continuously downmodulated in the DN4 thymocytes of control mice (Appendix A). In conclusion, the impaired DP thymocyte formation in *Pmca1^f/f^ pTα-Cre^+^* mice may have in part been attributed to the apoptosis of PMCA1-deficient DP thymocytes, while the DN thymocytes of these mice were much more resistant to apoptosis despite strongly elevated cytosolic Ca^2+^ levels. Since most DP thymocytes did not express Cre but were equally prone to apoptosis, it is, however, more likely that the apoptosis of DP thymocytes was a secondary consequence rather than the cause of the thymic atrophy in *Pmca1^f/f^ pTα-Cre^+^* mice.

## 3. Discussion

PMCA1 has long been viewed as a housekeeping enzyme due to the fact that its deletion is embryonically lethal in mice [35]. Most studies on PMCA functions in immune cells, where only PMCA1 and PMCA4 are present [40], have, therefore, focused on PMCA4. Undoubtedly, PMCA4 plays an important role in T cell activation by tuning TCR-induced SOCE [42,43,44,45]. PMCA1 has since been shown to be a key regulator promoting the exit from pluripotency in embryonic stem cells [37]. It was thus tempting to us to unravel the role of both isoforms in T cell development. 

Surprisingly, we found that PMCA4 was only expressed at very low levels compared to PMCA1 in mouse thymocytes and that its genetic deletion had no impact on the normal T cell development and global thymocyte Ca^2+^ homeostasis, but these results are in line with (1) a decline of *Pmca4* mRNA in DN3 thymocytes according to the ImmGen RNA expression database (www.immgen.org) [47], (2) our previous observation that PMCA4 was not required for normal B cell Ca^2+^ homeostasis and development [46], and (3) the observation that SOCE does not play a major role in the development of conventional αβ-T cells [8]. SOCE is, however, relevant for the development of regulatory T cells and other unconventional T cells [8]. The thymic numbers of Tregs, γδ-T cells or NKT cells were indeed slightly, but not significantly, lower in the *Pmca4^−/−^* mice compared to the control mice. The PMCA4 tuning of SOCE during the development of these cells is, thus, not as critical as for the activation of peripheral T cells. PMCA4 expression is more abundant in very early T progenitors suggesting that PMCA4 might play an unknown but probably redundant role in these progenitors. 

PMCA1 turned out to be the most prominent, if not the only, ATPase capable of expelling Ca^2+^ across the plasma membrane throughout thymocyte development including all the important selection phases. The deletion of *Pmca1* in DN2/3 thymocytes under the control of the pTα promoter resulted in a severe block of ISP and DP thymocyte formation with an accumulation of DN4 thymocytes that had dramatically increased basal intracellular Ca^2+^ levels. In line with this, we also found an almost complete lack of peripheral naïve T cells. It was puzzling, however, that these mice showed relatively high numbers of SP cells in the thymus and mature T cells with a highly activated phenotype in the spleen and lymph nodes. The low Cre-reporter expression and low basal Ca^2+^ levels in the DP and SP thymocytes, as well as the low abundance of mutated DNA and high PMCA1 protein expression in the peripheral T cells, led us to the conclusion that most of the T lineage cells beyond the DN stage in *Pmca1^f/f^ pTα-Cre^+^* mice derived from cells that escaped *Pmca1* deletion. One possible explanation for this would be a robust spontaneous proliferation [54] of a very limited number of surviving thymic emigrants. In addition, due to the phenotypic similarity of the high CD44 expression on SP thymocytes with the activated CD44^high^ phenotype of peripheral T cells, we assume that most of these cells were not the direct descendants of DP thymocytes but in fact were recirculating T cells homing back to the thymus. Rehoming to the thymus of primarily activated T cells has been observed under specific conditions, for instance during aging [55]. On the other hand, we also found that the very few existing DP thymocytes in the *Pmca1^f/f^ pTα-Cre^+^* mice were highly prone to apoptosis. Intuitively, these results would strongly suggest that the virtual absence of DP thymocytes is due to a fast removal of apoptotic cells. Indeed, ISP and DP are probably more sensitive than DN or mature SP thymocytes to Ca^2+^-induced apoptosis, e.g., through a switch towards less efficient anti-apoptotic Bcl2 isoforms [56]. PMCA1 may, therefore, set the threshold for survival, particularly in these sensitive cells, and the finding that PMCA1 is upregulated after the DN stage indicates a higher need of these cells for Ca^2+^ clearance. In this regard, PMCA1 may also be expected to be essential for survival during positive and negative selection. An evaluation of this hypothesis would require genetic approaches with the targeted deletion of *Pmca1* at later time points such as using a CD4-Cre to bypass the defects during the DN–DP transition. Apoptosis was not exclusively detected in actual PMCA1-deficient thymocytes, but it was prominent in all DP thymocytes. Assuming that even most of the DP thymocytes were wildtype, it is more likely to interpret DP thymocyte apoptosis as a result rather than the cause of the thymic atrophy, e.g., being induced by recirculating activated T cells [55]. These conclusions are altogether very speculative and can only be proven by tracking actual PMCA1-deficient, e.g., Cre reporter-positive, cells over time from newborn to adult mice in the thymus and peripheral lymphoid organs, which was far beyond the scope of this study. 

More importantly, several additional observations suggest that the severe impairment of thymocyte development in *Pmca1^f/f^ pTα-Cre^+^* mice becomes manifest before ISP and DP thymocytes formation. Specifically, the fact that DN4 not only survived high cytosolic Ca^2+^ levels but also accumulated at high numbers strongly argues for an intrinsic failure to fulfill the step of CD8 and CD4 upregulation. Moreover, when we initially cultured DN thymocytes on OP9-DL1 cells, which promotes ISP and DP development of wildtype cells [57], the DN thymocytes of *Pmca1^f/f^ pTα-Cre^+^* mice did not develop into ISP or DP at all (data not shown). The increased Ca^2+^ levels in both the DN3 and DN4 thymocytes of *Pmca1^f/f^ pTα-Cre^+^* mice correlated with signs of enhanced pre-TCR-associated signaling as indicated by an increased CD5 expression [50] and premature Notch1 downmodulation [51]. CD5 upregulation may indeed be controlled by Ca^2+^ signaling [58]. These phenotypes differ in several aspects from the phenotypes of mice with mutations targeting other Ca^2+^ handling or signaling molecules in early thymocytes. For example, most of these models, including mutations of Ca_V_1.2 and Ca_V_1.3 [18], TRPM7 [19] or Calcineurin B1 [15,20], resulted in a moderate to strong block at the DN3 stage related to impaired β-selection due to decreased Ca^2+^ signaling. Interestingly, mice with constitutive Calcineurin/NFAT activity in early thymocytes showed an impaired Rag recombinase expression and a strong developmental block also at the DN3 stage, suggesting Ca^2+^/Calcineurin/NFAT can act as a negative regulator of β-chain recombination [49]. Although specific functions of SERCA in T cells have not been addressed to our knowledge, a deficiency of SERCA2 and 3 in B cell development did lead to elevated cytosolic Ca^2+^ levels, reduced immunoglobulin gene recombination and B lymphopenia in mice and humans [59], which thus resembles the phenotype of hyperactive Calcineurin/NFAT in thymocytes [49]. We have not addressed β-chain recombination here, but impaired pre-TCR formation and pre-TCR signaling would lead to a developmental block before DN4 [48,49], rather than the accumulation of DN4 thymocytes as we observed in the *Pmca1^f/f^ pTα-Cre^+^* mice. The recombination of the β-chain was probably also finished before the pTα-driven *Pmca1* knockout in our mice. However, high Ca^2+^ levels in the DN4 thymocytes of *Pmca1^f/f^ pTα-Cre^+^* mice may transmit signals sufficient to overcome possible defects in pre-TCR signaling and promote DN4 differentiation.

The phenotype of *Pmca1^f/f^ pTα-Cre^+^* mice most likely resembles that of IP3 receptor 1–3 triple knockout mice in regard to a partial block at the DN4 and ISP stage [17]. Of course, this phenotype was due to an impaired Ca^2+^ release from the ER and not increased cytosolic Ca^2+^ levels. In addition, these mice developed T cell acute lymphoblastic leukemia (T-ALL) as a result of the impaired induction of the transcription factor TCF1 and prolonged oncogenic Notch1 signaling [17]. We did not observe any signs of T-ALL in the *Pmca1^f/f^ pTα-Cre^+^* mice which, rather, showed a pre-mature shutdown of Notch1 signaling. TCF1, however, is also important for determining the T lineage specification [60,61] that promotes β-selection and thymocyte maturation beyond [62], e.g., ISP formation [63]. As Ca^2+^ clearance by PMCA1 is required for a TCF7l1-mediated exit from pluripotency of embryonic stem cells [37], and all members of the TCF/LEF family including TCF1 are also involved in this process [64], it is tempting to speculate that very similar PMCA1 and Ca^2+^-controlled mechanisms may apply to the development of DP thymocytes. 

The data reported in this manuscript show that PMCA1 rather than PMCA4 plays a pivotal role during thymocyte development. Future work may further resolve whether a splice diversification of PMCA1 plays a role during this stepwise process. In fact, a developmental switch between type “b” and “a” splice variants, which differ in their cytoplasmic tails including the Ca^2+^/calmodulin binding regions [65], would likely affect the efficiency of calcium extrusion. Interestingly, splicing towards the supposedly more active PMCA1a variant during brain development was found to be promoted by CaMKIV [66], which in turn becomes upregulated during the DN to DP period [47,67]. 

We, therefore, propose that PMCA1 is not just a life-saving housekeeping enzyme but has an important function in the reprogramming for distinct developmental progression steps in thymocytes and probably other cell types. Particularly, it would be interesting to analyze specific functions of PMCA1 in the differentiation of mature T and B cells. This might also be relevant when considering PMCA-mediated Ca^2+^ signaling as a target for drug development.

## 4. Materials and Methods

### 4.1. Mice

*Pmca1^f/f^ pTα-Cre* mice on a C57BL/6J background were generated as previously described [33,46] by first intercrossing *Atp2b1^tm1a(KOMP)Wtsi^* mice obtained from the UC Davis Knockout Mouse Project (KOMP Repository collection, project ID CSD77635) with a FLPo deleter mouse (*B6.Cg-Tg(Pgk1-flpo)10Sykr/J,* The Jackson Laboratory) followed by intercrossing the resulting *Atp2b1^tm1c(KOMP)Wtsi^* (*Pmca1^f/+^*) mice with *pTα^iCre^* (herein referred to as *pTα-Cre*) knockin mice (*Ptcra^tm1(icre)Hjf^*) [68]. Mice already expressing a single mature T cell receptor were generated by intercrossing *Pmca1^f/f^ pTα-Cre* mice with OT-II transgenic mice [69]. *ROSA^mT/mG^* Cre reporter mice were generated by intercrossing *Pmca1^f/f^ pTα-Cre* mice with *B6.129(Cg)-Gt(ROSA)26Sor^tm4(ACTB-tdTomato,-EGFP)Luo^/J* which possess a cell membrane-targeted, two-color fluorescent Cre-reporter allele (The Jackson Laboratory, stock #007676, referred to here as *mT/mG*) [70]. *Pmca4^−/−^* mice on a C57BL6J background were generated by backcrossing FVB.129(Cg)-*Atp2b4^tm1Ges^*/Mmjax obtained from The Jackson Laboratory (MMRRC stock #36807) [35] with C57BL6/J mice (The Jackson Laboratory) for more than 8 generations. The mice were housed in specific-pathogen-free conditions and all the procedures were performed in accordance with the institutional guidelines for the health and care of experimental animals and were approved by the Landesverwaltungsamt Halle (representing the state of Saxony-Anhalt), Germany (Licence: 2-1181; IBZ-TWZ-01). Only generally healthy mice were included in the post mortem analyses.

### 4.2. Immune Cell Preparation

For the isolation of lymphocytes, the mice were sacrificed in a CO_2_ atmosphere, and the thymi, spleens and lymph nodes were dissected. Lymphocytes were dissolved from the tissues by crushing the organs through 40 µm nylon meshes and resuspended in cold PBS. Erythrocytes were lysed in 0.16 mM ammonium chloride solution. The cell numbers were determined by flow cytometry. 

The DN thymocytes were enriched by a magnetic depletion of CD4- and CD8-labeled cells using CD8a (Ly-2) and CD4 (L3T4) MicroBeads for mouse and an AutoMACS Pro Separator (Miltenyi Biotec, Bergisch Gladbach, Germany) according to the manufacturer’s guidelines. The purity of enriched DN thymocyte preparations was usually 75 to 90%. CD4^+^ T cells were enriched from secondary lymphoid organs using the CD4^+^ T cell Isolation Kit (Miltenyi Biotec, Bergisch Gladbach, Germany). The purity of CD4^+^ T cells was >90% for *Pmca1^f/f^ pTα-Cre^−^* and *Pmca1^f/+^ pTα-Cre^+^* mice. Due to variable purities for the *Pmca1^f/f^ pTα-Cre^+^* mice, only CD4^+^ T cell samples with purities ≥ 65% were used for the quantification. For the biochemical assays, cell pellets were frozen in liquid nitrogen and stored at −80 °C until further use.

### 4.3. Flow Cytometry

For the flow cytometric analysis of cell populations, single-cell suspensions were incubated with fluorochrome-conjugated antibodies for cell surface markers as noted in the figures and legends. A complete list of all the used antibodies is provided in Appendix A. NKT cells were labeled with a PE-conjugated mouse CD1d tetramer loaded with the α-GalCer analog PBS-57 (kindly provided by the NIH Tetramer Core Facility). Annexin V (Biolegend, Amsterdam, The Netherlands) staining was performed in an Annexin V staining buffer (i.e., 10 mM of Hepes pH7.4, 140 mM of NaCl, and 2.5 mM of CaCl_2_), and 7-aminoactinomycin D (7-AAD, Biomol, Hamburg, Germany) was added at 1 µg/mL 1–2 min before acquisition. For the intracellular detection of cytosolic proteins (i.e., active Caspase 3, and Notch1 intracellular domain) cells were first labeled for cell surface markers, then fixed and permeabilized using Fixation buffer and Intracellular staining Perm Wash Buffer (Biolegend), and stained with an intracellular antibody mix in Perm Wash Buffer. For nuclear antigens (i.e., Ki67, p53, and FoxP3), the cells were fixed, permeabilized and stained using the eBioscience^TM^ FoxP3/Transcription Factor Staining Buffer Set (Invitrogen, Carlsbad, CA, USA) according to the manufacturer’s recommendations. The stained cells were acquired on a BD FACSCanto II flow cytometer, and analyzed using the BD FACSDiva (BD Biosciences, Heidelberg, Germany) and FlowJo (version 10.6, Treestar, BD Biosciences) softwares. The gating of populations is shown in Appendix A and as representative 2D plots or histograms in the respective figures. The histogram y-axes scale was set to the modal option showing % of maximum count. 

### 4.4. Flow Cytometric Ratiometric Ca^2+^ Measurements

Ratiometric Ca^2+^ measurement in thymocyte suspensions was performed by flow cytometry which allowed for the simultaneous recording of different cell populations. For the Ca^2+^ analysis in DP, SP and total DN thymocytes, the cells were prelabeled with antibodies for CD4 (V450), CD8 (APC) and a mixture of dump antibodies for B220, CD11b, and Ter119 (APC/Cy) including a live/dead marker (Zombie NIR, Biolegend) in order to exclude B cells, myeloid cells and erythrocytes as well as dead cells. These labels did not interfere with the signals from the FL-1 and FL-3 channels used for the detection of the Ca^2+^ sensitive dyes Fluo-8 and FuraRed, respectively, and they did not alter the Ca^2+^ handling by the cells. For a better comparison of the DN thymocyte populations from *Pmca1^f/f^ pTα-Cre^+^* and control mice, we used DN thymocytes from *Pmca1^f/f^* mice enriched by MACS as described above. The DN thymocytes were labeled with anti-CD25 (BV421) and anti-CD44 (APC) antibodies to distinguish the DN1 to DN4 subpopulations and a dump mix as above including anti-CD4 and anti-CD8 (APC/Cy7) antibodies in order also to exclude the remaining DP and SP thymocytes. For the T cell receptor-specific stimulation, the cells were additionally prelabeled with 10 µg/ml of a hamster anti-mouse CD3ε antibody (clone 145-2C11). After surface labelling, the cells were then stained with 1.3 µg/mL of Fluo-8 and 2.7 µg/ml of FuraRed in RPMI1640 medium (all from ThermoFisher, Darmstadt, Germany) for 30 min at 37 °C/5% CO_2_ washed and resuspended in HBSS buffer (Biochrom AG, Berlin, Germany), which either contained 1.26 mM of Ca^2+^ or was free of Ca^2+^ and supplemented with 1 mM of EGTA. Before acquisition, the cells were preincubated at 37 °C/5% CO_2_ for 15 min. The Fluo-8 and FuraRed fluorescence were acquired on a BD FACSCanto II flow cytometer at 1 µl/s (recording several hundreds of cells per second). For the baseline measurement in the DN thymocyte populations fluorescence was recorded for three minutes. The TCR-induced Ca^2+^ influx was measured after one minute of baseline recording followed by the addition of 10 µg/mL of an anti-hamster F(ab’)_2_ in order to crosslink the TCR molecules on anti-CD3 labeled cells and then recorded for further 10 min. Ca^2+^ release from the ER was measured for 10 min in a Ca^2+^-free buffer after a one minute baseline recording and the addition of 1 µM of Thapsigargin (Tg) to specifically and irreversibly block SERCA.

The gates were set using the FSC, SSC, APC, APC-Cy and V450 (or BV421) signals to distinguish the viable cell subsets and the Fluo-8/FuraRed ratio was calculated using the FlowJo software (version 7.6, Treestar). The kinetics data were generated with the FlowJo Kinetics Tool and then processed and plotted using Excel (version 14.0, Microsoft). Each time point represents the mean ratio of cells acquired in consecutive intervals of two to three seconds. The basal cytosolic Ca^2+^ level in DN subpopulations was quantified as the mean Fluo-8/FuraRed ratio ± SD measured for 3 min. For a comparison of the basal and induced Ca^2+^ levels in the total DN, DP and SP thymocytes, the data were expressed as the fold change above (or below) the basal Ca^2+^ level of the respective wildtype population. To this end, the area under the curve (AUC) of the Fluo-8/FuraRed kinetics graph was calculated and divided by the respective time interval t. The data were then normalized by dividing each AUC/t value by the AUC/t value of the basal Ca^2+^ level of the respective wildtype population. 

### 4.5. Preparation of Protein Extracts and Western Blotting

Cell pellets were homogenized by incubation at 4 °C for 30 min in a Triton-homogenization buffer (i.e., 20 mM of Tris pH 7.5, 150 mM of NaCl, 1% Triton-X-100, 2 mM of MgCl_2_, 750 U/mL of Benzonase (Sigma, Darmstadt, Germany), and protease inhibitors (cOmplete^TM^ Protease Inhibitor cocktail tablets, Roche, Mannheim, Germany), centrifuged at 15,000× *g* for 20 min, and the resulting supernatants were used for the analysis. The protein content in the supernatant was determined using a bicinchoninic acid (BCA) kit (Pierce, Rockford, IL, USA) according to the manufacturer’s instructions. The samples were denatured (at 95 °C, for 5 min) in a sample loading buffer (RotiLoad I, Roth, Karlsruhe, Germany) at concentrations of approximately 0.8–1.5 µg/µL and run on SERVA*Gel*^TM^ TG PRiME^TM^ 4–20% Gels (Serva, Heidelberg, Germany). The proteins were then immunoblotted according to the standard procedures and the immunoreactivity was detected using an ECL Imager (GeneGnome XRQ, Syngene, Cambridge, UK). The quantification of band intensities was performed using ImageJ, and the OD values were normalized to either ERK1/2 or β-actin as the loading controls.

### 4.6. Evaluation of Cre Activity in Peripheral CD4^+^ T Cells by Genomic PCR

A scheme of the genomic PCR products of wildtype and mutated *Pmca1* alleles is shown in Appendix A. The genomic DNA was prepared using the Nucleo Spin Tissue Kit according to the manufacturer’s instructions (Macherey Nagel, Düren, Germany). Equal ng amounts (140 ng) of genomic DNA were used for all PCRs. Based on the PCR strategy given by the KOMP repository (www.komp.org; KOMP PCR Design), the PCR was performed using the following primers: CSD-Atp2b1-F (5′-CTAGGCATATAATGGGCGAAGCAGC-3′ located in the 5′-arm of the knockout construct) and CSD-Atp2b1-R (5′-CCAGATGGCATGTCTCAACATCAGC-3′ located in the 3′-arm), using a PCR synthesis step of 2 min to allow for the synthesis of long amplicons. A 1434 bp product originated from a wildtype allele, while the floxed alleles, but not cut by Cre, resulted in a PCR product of 1639 bp. An amplicon of 750 bp is indicative of a floxed/floxed allele after a Cre cut. The amplification products were separated by gel electrophoresis.

### 4.7. Examination of Relative Levels of Pmca1 and Pmca4 mRNA Expression

The total RNA from mouse cells was isolated using the NucleoSpin RNA Kit according to the manufacturer’s instructions (Macherey Nagel). The total RNA from mouse tissues was isolated using peqGold TriFast^TM^ (peqlab, Erlangen, Germany) according to the manufacturer’s instructions. The analysis was performed according to a method published originally by Okunade et al. [35] and recently adapted by our group [46]. This strategy is based on first simultaneously amplifying *Pmca1* and *Pmca4* cDNA fragments, followed by a gene-specific restriction digest of the PCR products. The first strand cDNA was prepared using the RevertAid H Minus First Strand cDNA Synthesis Kit (Thermo Fisher Scientific, Baltics UAB, Vilnius, Lithuania). Briefly, 375 ng of total RNA were reverse transcribed using random primers, followed by a PCR using the primers published by Okunade et al. [35]. These primers allow for the efficient amplification of a 1 kb region of *Pmca1* and *4,* but not *Pmca2* and *3*. Restriction digests of the PCR products were performed with either ClaI (C) or BglI (B) or both (CB) (FastDigest enzymes, Thermo Fisher Scientific) or neither (uncut, U), yielding fragments of 400 bp and 600 bp characteristic for *Pmca1* (ClaI cut) or two ~500 bp fragments for *Pmca4* (BglI cut) as shown in the scheme in Figure 1E.

### 4.8. Statistical Analysis

The data are reported as the mean ± SD or SEM, as indicated. Differences were tested by an unpaired Student’s *t*-test for a comparison of two groups and a one-way ANOVA for the comparison between three groups, using Excel or GraphPad Prism v8, respectively. The difference between two data sets was considered significant for *p* values < 0.05.

## Figures and Tables

**Figure 1 ijms-24-01442-f001:**
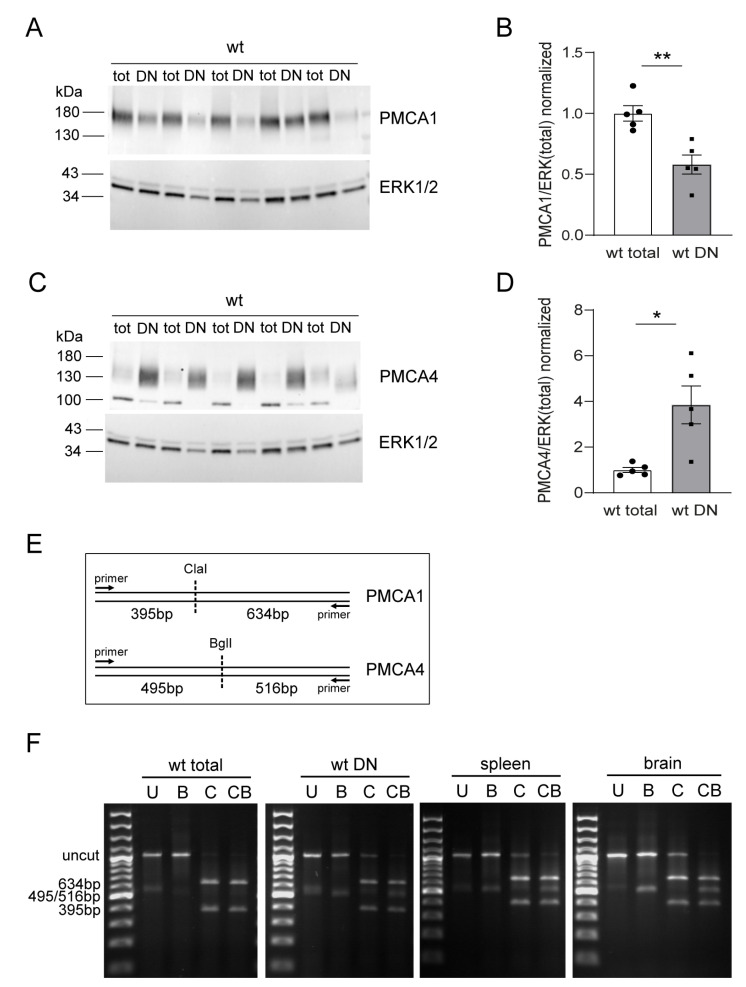
PMCA1 is the dominant PMCA isoform during T cell development. (**A**–**D**) Western blot analysis of PMCA1 (**A**,**B**) and PMCA4 (**C**,**D**) protein expression in lysates (15–30 µg/lane) from total thymocyte and DN thymocyte pellets obtained from 5 wildtype (wt) mice. ERK1/2 was used as the loading control. The molecular weight of PMCA1 appears higher than the expected weight of ~135 kDa, probably due to an unspecific aggregation. PMCA protein band intensities were quantified and divided by the respective ERK1/2 intensity. Bar graphs (**B**,**D**) show the mean relative protein level ± SEM normalized to the mean of total thymocytes. * *p <* 0.05, ** *p <* 0.01, by an unpaired Student’s *t*-test. (**E**,**F**) mRNA expression levels of *Pmca1* and *Pmca4* in total versus DN thymocytes by RT-PCR using a relative PCR strategy. (**E**) Scheme, depicting the size of fragments after a restriction digest of PCR products with ClaI (C), BglI (B), both (CB) or neither (U). (**F**) Representative agarose gels showing *Pmca1* and *Pmca4* specific restriction fragments in indicated thymocyte or control samples. Mouse spleen and brain tissue served as controls. The image is representative of three independent biological samples.

**Figure 2 ijms-24-01442-f002:**
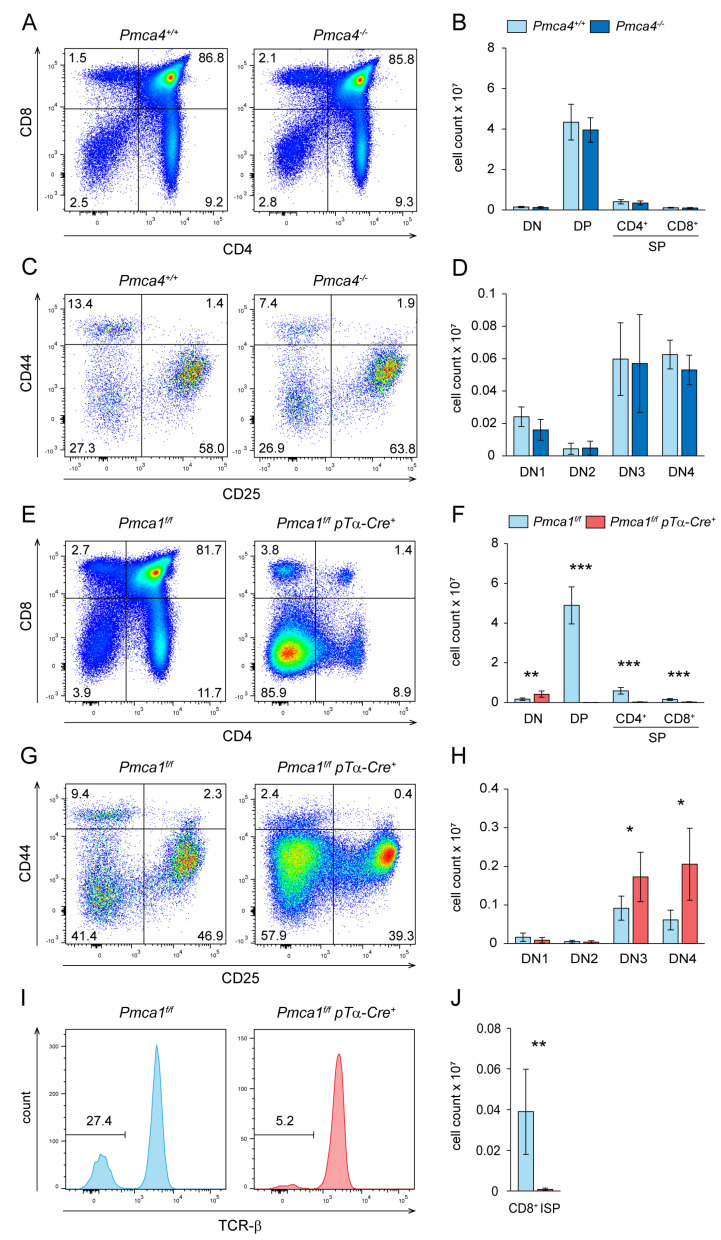
T cell development is normal in mice with PMCA4 deficiency but almost completely blocked in mice with a conditional deletion of the *Pmca1* gene. Thymocytes of *Pmca4^+/+^* control and *Pmca4^−/−^* mice (**A**–**D**) or *Pmca1^f/f^ pTα-Cre^−^* control and *Pmca1^f/f^ pTα-Cre^+^* mice (**E**–**J**), respectively, were analyzed by flow cytometry. (**A**) Representative FACS plots of CD4 and CD8 stained thymocytes and (**B**) quantification of the total numbers of DN, DP, CD4^+^ SP and CD8^+^ SP thymocytes obtained from *Pmca4^+/+^ and Pmca4^−/−^* mice. (**C**) Representative CD44/CD25 FACS plots of gated DN thymocytes to discriminate CD44^high^CD25^−^ DN1, CD44^high^CD25^+^ DN2, CD44^low^CD25^+^ DN3 and CD44^low/−^CD25^−^ DN4 thymocytes and (**D**) quantification of total numbers of these populations obtained from *Pmca4^+/+^ and Pmca4^−/−^* mice. (**E**) Representative CD4^+^CD8^+^ FACS plots of thymocytes and (**F**) quantification of total numbers of DN, DP and SP thymocytes obtained from *Pmca1^f/f^ and Pmca1^f/f^ pTα-Cre^+^* mice. (**G**) Representative CD44/CD25 FACS plots of DN thymocytes and (**H**) quantification of total numbers of DN1 to DN4 thymocytes in *Pmca1^f/f^* and *Pmca1^f/f^ pTα-Cre^+^* mice. (**I**) Representative FACS histograms of TCR-β expression on CD8^+^ SP thymocytes to discriminate TCR-β^−^ CD8^+^ ISP from mature TCR-β^+^ CD8^+^ SP thymocytes and (**J**) quantification of the absolute ISP cell numbers in thymi from *Pmca1^f/f^* and *Pmca1^f/f^ pTα-Cre^+^* mice. The numbers in plot quadrants and histograms indicate the proportion of respective populations within the indicated gates. All diagrams show the mean absolute cell numbers ± SD of the indicated thymocyte populations combined from 5 experiments (**B**), 4 experiments (**D**) and 6 experiments (**F**,**H**,**J**), respectively. * *p* < 0.05, ** *p* < 0.01, *** *p* < 0.001, all by an unpaired two-tailed *t*-test. Gating of singlet thymocytes is shown in Appendix A.

**Figure 3 ijms-24-01442-f003:**
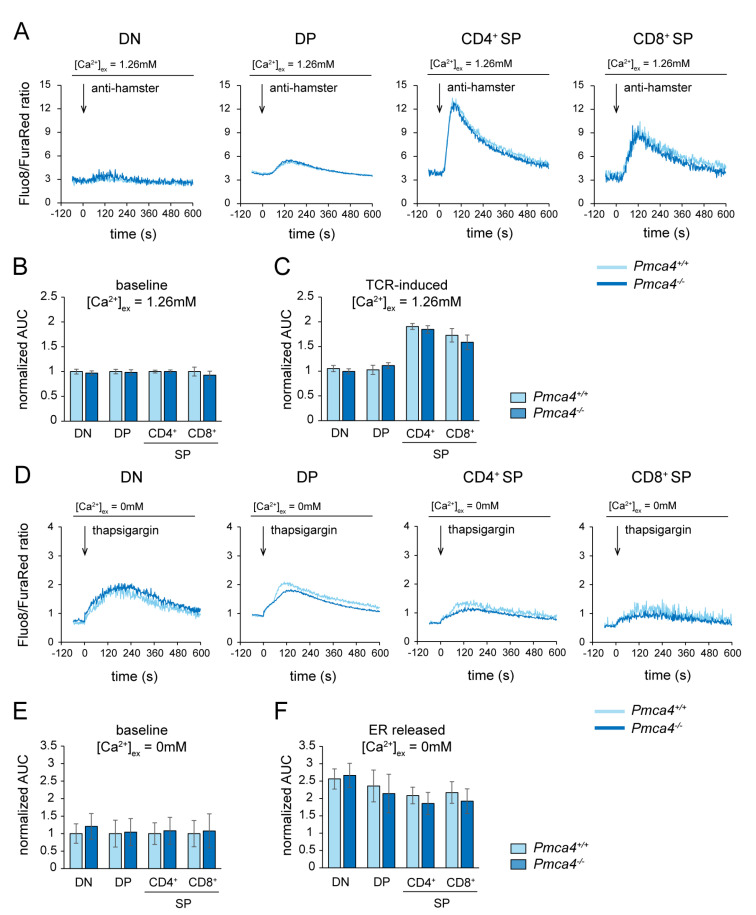
PMCA4 is not required for global intracellular Ca^2+^ homeostasis during T cell development. (**A**–**C**) Flow cytometric ratiometric measurements of basal Ca^2+^ level and TCR-induced Ca^2+^ kinetics in Fluo-8 and FuraRed labeled thymocytes of *Pmca4^+/+^ and Pmca4^−/−^* mice in a Ca^2+^-containing buffer. The representative kinetic graphs (**A**) show changes of the Fluo-8/FuraRed ratio in anti-CD3 labeled DN, DP and SP thymocytes for 10 min upon the addition of a CD3 crosslinker (anti-hamster) after a one minute acquisition of the baseline level. For the quantification of basal (**B**) and TCR-induced (**C**) Ca^2+^ levels, the AUC values were normalized to the corresponding basal value of the control as described in the methods section. The bar graphs show the mean normalized AUC values ± SD obtained from 3 experiments. (**D**–**F**) Flow cytometric ratiometric measurements of the basal Ca^2+^ level and kinetics of Ca^2+^ release from the ER upon the blockage of SERCA in the thymocytes of *Pmca4^+/+^ and Pmca4^−/−^* mice in a Ca^2+^-free buffer. The kinetic graphs (**D**) show changes of the Fluo-8/FuraRed ratio in DN, DP and SP thymocytes for 10 min upon the addition of thapsigargin following a one minute baseline acquisition. For the quantification of basal Ca^2+^ levels in the Ca^2+^-free buffer (**E**) and ER Ca^2+^ release (**F**), the AUC values were normalized to the corresponding basal value of the control. The bar graphs show the mean normalized AUC values ± SD obtained from 5 experiments.

**Figure 4 ijms-24-01442-f004:**
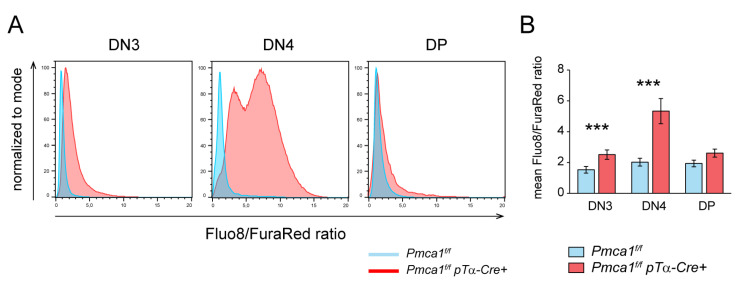
PMCA1 keeps cytosolic Ca^2+^ levels low during T cell development. (**A**,**B**) Flow cytometric ratiometric measurements of basal cytosolic Ca^2+^ level in Fluo-8- and FuraRed-labeled thymocytes of *Pmca1^f/f^* and *Pmca1^f/f^ pTα-Cre^+^* mice in a Ca^2+^-containing buffer. The Fluo-8/FuraRed fluorescence intensity ratios in the DN3, DN4 and DP thymocytes, respectively, are shown in the representative histograms (**A**) and were quantified as the mean Fluo-8/FuraRed ratio ± SD acquired over 3 min from 5 experiments (**B**). *** *p* < 0.001, all by an unpaired two-tailed *t*-test.

**Figure 5 ijms-24-01442-f005:**
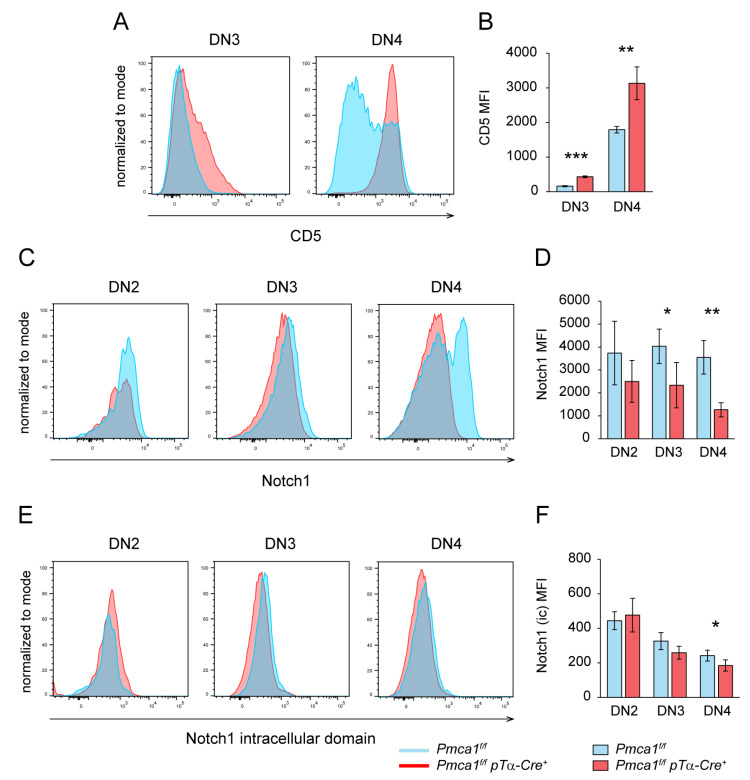
DN thymocytes of *Pmca1^f/f^ pTα-Cre^+^* mice show pre-mature signaling. (**A**,**B**) FACS analysis of the CD5 surface expression on DN thymocytes of *Pmca1^f/f^* and *Pmca1^f/f^ pTα-Cre^+^* mice shown as representative histograms (**A**) of the CD5 fluorescence intensity on DN3 and DN4 thymocytes, respectively, and quantified (**B**) as the mean MFI ± SD combined from 3 experiments. (**C**,**D**) FACS analysis of the Notch1 surface expression in DN thymocytes of *Pmca1^f/f^* and *Pmca1^f/f^ pTα-Cre^+^* mice. Representative histograms (**C**) of Notch1 are shown for DN2, DN3 and DN4 thymocytes, respectively. The bar graph (**D**) shows the mean MFI ± SD of Notch1 surface staining in these populations combined from 3 experiments. (**E**,**F**) Intracellular FACS analysis of the cleaved cytosolic Notch1 domain in DN thymocytes of *Pmca1^f/f^* and *Pmca1^f/f^ pTα-Cre^+^* mice. Histograms of the cleaved Notch1 fluorescence intensities are shown for DN2, DN3 and DN4 thymocytes, respectively (**E**). The bar graph (**F**) shows the mean MFI ± SD combined from 3 experiments. * *p* < 0.05, ** *p* < 0.01, *** *p* < 0.001, all by an unpaired two-tailed *t*-test.

**Figure 6 ijms-24-01442-f006:**
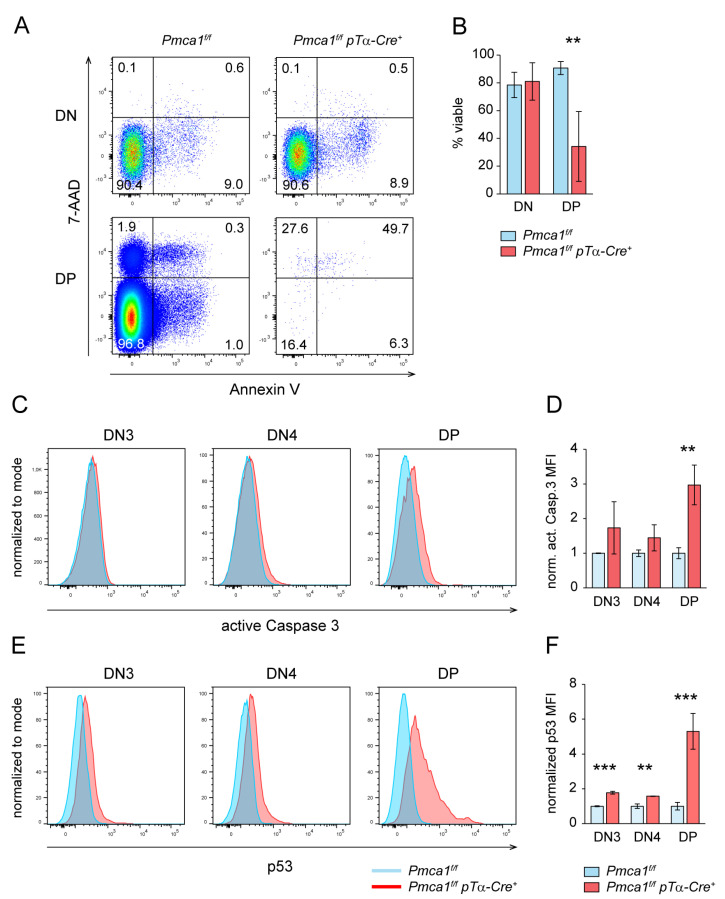
DP but not DN thymocytes of *Pmca1^f/f^ pTα-Cre^+^* mice are strongly prone to apoptosis. (**A**,**B**) Ex vivo Annexin V and 7-AAD staining of DN and DP thymocytes of *Pmca1^f/f^* and *Pmca1^f/f^ pTα-Cre^+^* mice. The representative FACS plots (**A**) show Annexin V^+^/7-AAD^−^ early apoptotic, Annexin V^+^/7-AAD^+^ late apoptotic, Annexin V^−^/7-AAD^+^ dead and Annexin V^−^/7-AAD^−^ viable cells within the DN and DP thymocyte populations, respectively. The numbers in plot quadrants indicate the proportion of respective populations. Viability is summarized in the bar graph (**B**) as the mean proportion of viable Annexin V^−^/7-AAD^−^ double negative cells ± SD obtained from 2 experiments. (**C**,**D**) Intracellular FACS staining of active Caspase 3 in DN and DP thymocytes of *Pmca1^f/f^* and *Pmca1^f/f^ pTα-Cre^+^* mice. The representative histograms (**C**) show the fluorescence intensity of active Caspase 3 staining in DN3, DN4 and DP thymocytes, respectively. For quantification, the MFI values for each population were normalized to the MFI of the corresponding population of the control in each experiment. The bar graph (**B**) shows the mean normalized MFI ± SD obtained from 2 experiments. (**E**,**F**) Intracellular FACS staining of p53 expression in DN and DP thymocytes of *Pmca1^f/f^* and *Pmca1^f/f^ pTα-Cre^+^* mice. The representative histograms (**E**) show the fluorescence intensity of p53 staining in DN3, DN4 and DP thymocytes, respectively. For quantification, the MFI values were normalized as in D. The bar graph (**F**) shows the mean normalized MFI ± SD obtained from 2 experiments. ** *p* < 0.01, *** *p* < 0.001, all by an unpaired two-tailed *t*-test.

## Data Availability

The data presented in this study are available from the corresponding author upon reasonable request.

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
