# Peer review of "Ca2+ Homeostasis by Plasma Membrane Ca2+ ATPase (PMCA) 1 Is Essential for the Development of DP Thymocytes"

_ijms, 2023, doi:10.3390/ijms24021442_

Round 1

Author Response

Dear reviewer, please see the attachment. We provide the responses to all three reviewers as one file. Part of our response refers to the comments of more than one reviewer, where indicated.    

Reviewer 2 Report

This report by Beckmann et al. is very impressive and the distinction between PMCA1 and 4 for thymocyte development could be of high scientific interest. The authors provide convincing evidence that PMCA1 is essential for  thymocyte development but not PMCA4 which is surprising since both PMCA isoforms are usually ubiquitously expressed. On the other hand, this difference in a way is reminiscent of the role of PMCA1 during brain development, where the spliced isoform PMCA1a is essential. Since the splicing of PMCA1a is under the control of the calmodulin dependent protein kinase IV (CaMKIV; e.g. see Krebs Curr. Top. Med. Chem. 21,1121,2021) it would be very interesting if during T cell development the actual PMCA isoform would be PMCA1a, the more since the crucial importance of PMCA1 occurs between the DN and the DP stage of thymocyte development, exactly when CaMKIV expression is significantly increased (Krebs et al. BBRC 241, 383-389, 1997; Racioppi and Means Trends Immunol. 29, 600-607, 2008). Such an aspect could be discussed and could be important for future experiments.

Minor points:

1) In Fig. 1A PMCA1 is indicated with a MW of ca. 180 kDa, but the size of PMCA1 should be around 135 kDa. Any explanation?

2) In Fig 1A it is shown that PMCA1 is low in DN cells, but in lane 8 it is high, why?

Author Response

(The authors gave the same response as above.)

Reviewer 3 Report

The manuscript entitled, “Ca2+ homeostasis by plasma membrane Ca2+ ATPase (PMCA) 1 is essential for the development of DP thymocytes” provides important insight into the roles of both PMCA1 and cytosolic Ca2+ content in the development of conventional T cells. Briefly, it is established that PMCA1 is the major PM Ca2+ ATPase in thymocytes, particularly in DP thymocytes after a developmental drop in the expression of PMCA4. They further establish that loss of PMCA1 leads to a near total loss of DP thymocytes, with those few detectable cells representing failure to delete PMCA1 rather than development of small numbers of PMCA1-null cells. Finally, they establish that loss of PMCA1 leads to signs of early maturation of DN3 and DN4 thymocytes and then apoptosis in DP thymocytes (based on expression of caspase 3). Overall, I think that this is an important paper; I have only one question that I think should be addressed.

Comments:

As I understand the findings of the paper, the majority of the DP thymocytes that form in Pmca1f/f pTα-Cre+ mice escaped deletion of PMCA1. However, the data in figure 6 shows that DP thymocytes in these mice are prone to apoptosis. The concept is credible, but please clarify how this experiment was performed such that actual PMCA1-/- cells could be studied.

Author Response

(The authors gave the same response as above.)

Round 2

Reviewer 1 Report

Now, I agree with publication of the revised manuscript. Thanks for adding the CD44 data of the SP thymocytes. I can also understand that evaluation of the periphery can be left for future studies and am thankful for the more vage discussion on this point.

Very minor, but I think you should add a '2' in line 536 'antiapoptotic Bcl2 isoforms'. Bcl can be anything  originally found in B-cell lymphoma and the term is also still used within Bcl3 and Bcl6.